# Integrating AlphaFold2 models and clinical data to improve the assessment of Short Linear Motifs (SLiMs) and their variants' pathogenicity

Franco Gino Brunello[1]*, Lorenzo Erra[1], Juan Nicola[2], Marcelo Adrián Martí[1]*

1 Departamento de Química Biológica, Facultad de Ciencias Exactas y Naturales, Universidad de Buenos Aires (FCEyN-UBA) e Instituto de Química Biológica de la Facultad de Ciencias Exactas y Naturales (IQUIBICEN) CONICET, Pabellón 2 de Ciudad Universitaria, Ciudad de Buenos Aires, Argentina, 2 Departamento de Bioquímica Clínica, Facultad de Ciencias Químicas, Universidad Nacional de Córdoba. Centro de Investigaciones en Bioquímica Clínica e Inmunología - Consejo Nacional de Investigaciones Científicas y Técnicas (CIBICI-CONICET), Córdoba, Argentina

* fgbrunello@gmail.com (FGB); marti.marcelo@gmail.com (MAM)

## Abstract

Short Linear Motifs (SLiMs) are protein functionally relevant regions that mediate reversible protein-protein interactions. Variants that disrupt SLiMs can lead to numerous Mendelian diseases. Although various bioinformatic tools have been developed to identify SLiMs, most suffer from low specificity. In our previous work, we demonstrated that integrating sequence variant information with structural analysis can enhance the prediction of true functional SLiMs while simultaneously generating tolerance matrices that indicate whether each of the 19 possible single amino acid substitutions (SASs) is tolerated. However, the scarcity of representative crystallographic structures of SLiM-receptor complexes posed a significant limitation. In this study, we demonstrate that these interactions can be modeled using AlphaFold2 (AF2) to generate high-quality structures that serve as input for our MotSASi method. These AF2-derived structures show robust performance, both in reproducing known structures deposited in the Protein Data Bank (PDB) and in reflecting the deleterious effects of known sequence variants. This updated version of MotSASi expands the repertoire of high-confidence predicted SLiMs and provides a comprehensive catalog of variants located within SLiMs, along with their respective deleteriousness assessments. When compared to AlphaMissense, MotSASi demonstrates superior performance in predicting variant deleteriousness. By contributing to the accurate identification and interpretation of variants, this work aligns with ACMG/AMP standards and aims to improve diagnostic rates in clinical genomics.

**Data availability statement:** The data supporting the findings of this study, along with the custom Python 3 scripts, are available at: https://github.com/fginob1/MotSASi2.0 DOI: 10.5281/zenodo.15189546.

**Funding:** The author(s) received no specific funding for this work.

**Competing interests:** The authors have declared that no competing interests exist.

## Author summary

Proteins interact with each other in highly specific ways to carry out vital biological functions. Short Linear Motifs (SLiMs) are small regions within proteins that mediate many of these reversible interactions. Changes in SLiMs can disrupt these interactions and lead to severe genetic disorders. Identifying SLiMs accurately has been a longstanding challenge, as many computational tools suffer from low specificity. Previously, we developed a method, MotSASi, that combines sequence variation data and structural analysis to improve SLiM prediction and assess the impact of single amino acid substitutions (SASs). However, the lack of available structural data limited its application. In this study, we demonstrate that structures generated using AlphaFold2 (AF2) can overcome this limitation. These high-quality AF2 models reliably reproduce known structures and capture the harmful effects of sequence variations. By integrating AF2 models, the updated MotSASi method identifies more high-confidence SLiMs and provides detailed assessments of the variants within them. MotSASi outperforms existing tools, such as AlphaMissense, in predicting the impact of genetic variants, offering insights aligned with clinical standards. This advancement can aid in understanding disease mechanisms and improving genetic diagnostics in clinical genomics.

## Introduction

Short Linear Motifs (SLiMs) are short stretches of amino acids found in protein sequences, often defined by regular expressions, and typically located in disordered regions of proteins or exposed flexible loops. These characteristics allow SLiMs to interact with their domain-binding partners reversibly and transiently, contributing to proper biological function of the protein host. SLiMs are crucial for normal cellular physiology, participating in various processes such as cell cycle, vesicle trafficking, cytoskeleton dynamics, innate immunity, and protein and RNA degradation systems, all of which play key roles in regulating cellular decision-making [1]. Given their significant biological roles, their identification and characterization are essential for understanding molecular cell biology. However, due to the difficulties and limitations of their experimental identification, even considering relatively novel experimental medium- and high-throughput approaches [2,3], only a small fraction of SLiMs have been deeply studied to date. In contrast, since most human (as well as other species) protein sequences are known, thousands of SLiMs have been bioinformatically predicted and await further scrutiny [4].

SLiM prediction and analysis are commonly approached using diverse bioinformatic tools [5–12], among which the universally recognized Eukaryotic Linear Motif (ELM) resource stands out. ELM is a comprehensive SLiM database and motif prediction tool [5]. In the latter role, it leverages information on sequence, structure, function, localization, evolutionary conservation, and interaction context to evaluate the confidence of motif-mediated protein complexes. Despite these advancements,

SLiM prediction remains impaired by very low specificity, leading to a high rate of false positives. Our estimation, based on the results of our previous work, suggests that the false-positive rate could exceed 80% (5,056 high-confidence potential motifs containing clinically relevant variants, compared to 29,428 total sequence hits across the human proteome that also carry clinically relevant variants, identified using regular expressions for 20 motif classes). This estimate is consistent with the findings of Idrees et al., based on the ratio of true binders to non-binders in yeast two-hybrid systems using randomly generated peptides [13].

This issue arises for several reasons, including the vastness of the search space (e.g., the whole human proteome) and the degenerate nature of many SLiM regular expressions, which makes their occurrence by chance, highly probable. Furthermore, the fact that many SLiM classes in the ELM database contain very few validated instances (i.e., true positives) exacerbates the problem. In this context, downstream filtering strategies that improve SLiM prediction confidence are essential, which was the focus of our previous work, where we presented the Motif-occurring Single Amino acid Substitution information (MotSASI) method [14].

MotSASi leverages the structural information of the SLiM-receptor complexes to analyze the effects of single amino acid substitutions (SAS) within the motif, employing the FoldX software [15,16]. This analysis is further integrated with the clinical significance and population allele frequency of motif located missense variants [17,18]. The underlying premise is that non-functional SLiMs identified by chance through regular expression matching (false positives), exhibit a variation pattern—pathogenic (non-tolerant) versus benign (tolerant)—that significantly differs from that of functional SLiMs (true positives). The initial implementation yielded promising results, achieving significant enrichment of true positives, while also filtering out a substantial number of false positives. However, the algorithm was constrained to classifying only candidates with available crystallographic structures of motif-domain complexes, thereby limiting the analysis to just 10–20% of all motif matches in the proteome. MotSASi has been recognized as a valuable tool for characterizing intrinsically disordered regions, such as amyloid-beta, and has been well received by the research community [19–21].

Since the deployment of our first version, gnomAD and ClinVar databases have continued to expand, and the increasing availability of genomes and exomes deposited in gnomAD suggests significant potential for improved outcomes. More importantly, recent advances in machine learning approaches for protein structure prediction, such as AlphaFold, RoseTTAFold or ESMFold, have completely changed the protein structure landscape [22,23]. These developments promise to bridge the gap that previously prevented the study of many more motif classes.

AlphaFold2 (AF2), the state-of-the-art AI system developed by Google DeepMind, computationally predicts protein structures with unprecedented accuracy and speed [24]. The system takes a query sequence as input and predicts first residue-residue contacts, and then the resulting structure by combining the construction of multiple sequence alignments. More recently, the user interface has been enhanced to support the incorporation of multiple simultaneous query chains, seamlessly aligning with our objective of modeling motif-domain interaction pairs [25]. Moreover, recent studies have shown that AF2's has already surpassed the ability of other docking programs in predicting and ranking peptide-protein complexes, and that it allows identifying those peptides that might have the highest binding affinity and their structure [26]. Importantly, AF2 has also been successfully applied to study domain–motif interactions in the context of neurodevelopmental disorders, although predictive accuracy was shown to decrease when full-length proteins containing the motif were used as input, highlighting the need for tailored fragmentation strategies [27].

In the present study, we hypothesized that AlphaFold's structural prediction capabilities would allow the generation of models for all human SLiMs, in complex with their domain-binding partners for those motif classes lacking representative crystallographic structures. Furthermore, we propose that integrating the resulting SAS structural tolerance matrix with the increasingly available allele frequency and clinical variant data would lead to: i) a significant increase in the coverage of potential true SLiMs in the human proteome, and ii) an improved method for predicting the pathogenicity of missense variants located within SLiMs.

## Materials and methods

### ELM motif classes and regular expressions

Motif classes from the ELM resource were selected based on the following criteria: they had to have a fixed length, at least one experimentally validated instance in humans (i.e., a true positive), and a minimum of five pathogenic or benign variants. Classes associated with post-translational modifications were excluded. The selected classes were subdivided into those with a crystallographic structure available in the Protein Data Bank (PDB), and those without. In total, 24 motif classes from the former group and 27 from the latter were studied. A subset of the former group (17/24) was used in the AF2 motif-domain complex structure prediction validation procedure. The defining regular expressions for these classes were written following the nomenclature specified in the ELM database. Specific ELM databases utilized in this work include *elm_instances.tsv*, *elm_classes.tsv*, *elm_interactions.tsv*, and *elm_interaction_domains.tsv* (last accessed version corresponds to September 2024).

### SLiMs identification in the Human proteome

The whole human proteome was defined as the 20,435 canonical protein sequences with Reviewed status (i.e., Swiss-Prot) in the UniProt database (as of September 2024) [28]. SLiM regular expression matches were identified using an in-house-developed Python script that scanned the entire sequences, allowing multiple hits per sequence. Matches were defined as occurrences of the SLiM regular expression within the analyzed protein, characterized by the specific sequence, its position, and its length. The complete set of all such matches identified across the human proteome for a given motif class is called the "Initial Set".

### Identification of naturally occurring variants within each motif

Clinically relevant missense variants involved in SLiM analysis were extracted from the ClinVar (FullRelease, September 2024) and gnomAD (v4.0 - Exomes, November 2023) databases. Only variants classified in ClinVar as Pathogenic, Pathogenic/Likely Pathogenic, Likely Pathogenic, Benign, Benign/Likely Benign, or Likely Benign were considered. Clinical significance and the number of corresponding supporting submissions were recorded for these variants. Benign missense variants exceeding a predefined allele frequency threshold were retrieved from gnomAD. For genes with at least 10 pathogenic variants, the allele frequency threshold was determined based on the most frequent pathogenic variant reported in ClinVar. For genes with fewer than 10 pathogenic variants, thresholds based on autosomal recessive (AR) and autosomal dominant (AD) inheritance models were applied ($10^{-4.1}$ and $10^{-4.28}$, respectively), informed by the respective allele frequency distributions of pathogenic and benign variants in ClinVar. Finally, for genes not yet associated with any disease, a fixed threshold of $10^{-4.1}$ was applied. The threshold values varied from 0.7637 to 0.000015, and are those commonly used in our group for variant annotation, following ACMG criteria [29].

### Structure data set

The crystallographic structures associated with motif instances of ELM classes were retrieved from the PDB. For ELM classes without associated crystallographic structures, 10 predictive models were generated using AlphaFold2, with the motif and domain sequences of experimentally validated human instances (i.e., true positives) from the ELM database as input. These models were computed using two different settings for the number of recycles: 24 (five models per class) and 72 (five models per class). The runtime estimation for running AF2 depends on the sequence length. Our calculations show that processing a single residue takes $0.89 \pm 0.14$ seconds per residue for the first model and $0.83 \pm 0.16$ seconds per residue for the remaining four models. These estimations were obtained using the following hardware specifications: RYZEN 9 5900X processor (24 cores), 62 GB RAM, 4 TB storage, and an NVIDIA GeForce RTX 3050 GPU.

## FoldX free-energy change calculations

During the pipeline execution and SAS matrix generation, the stability Gibbs free-energy change ($\triangle\triangle$G) for each missense variant in the SLiM was calculated using FoldX, using the corresponding motif-domain complex structure (derived either from PDB or AF2). Calculations were performed using the PositionScan command. The resulting $\triangle\triangle$G values for each SAS in each motif class are presented as structural stability free-energy substitution matrices. When multiple structures were available for a given complex, the matrix values represent the average $\triangle\triangle$G across all structures.

To determine the SLIM-receptor interaction energy, establish the corresponding threshold, and further filter the AF2-generated models, we used the FoldX AnalyseComplex command across the different AF2 models.

## Clinical significance, population allele frequency, and free-energy change conversion into confidence scores

Missense variants classified as "Benign", "Benign/Likely benign", "Likely benign", "Pathogenic", "Pathogenic/Likely pathogenic" or "Likely pathogenic" were obtained from ClinVar. The clinical significance of these variants was converted into confidence scores, assigning 1 point per review status star and 0.1 points per supporting submission. For example, a variant classified as "Pathogenic" in ClinVar with a two-star review status and three supporting submissions would receive a confidence score of +2.3. Conversely, a variant classified as "Benign" with a one-star review status and one supporting submission would receive a confidence score of -1.1.

Population allele frequencies from gnomAD and FoldX stability free-energy change values were also transformed into confidence scores using the respective equations described in Figure B in S1 Text.

## Conservation score calculation

Multiple sequence alignments (MSAs) were constructed using sequences from the UniRef50 cluster associated with each match-containing protein. The alignments were performed using MAFFT software and subsequently used to calculate the motif conservation score, determined through the Jensen-Shannon Divergence algorithm [30,31].

## Secondary structure prediction

Protein secondary structure prediction was performed using the Scratch software (an ensemble prediction method) with the full sequence provided as input [32]. The specific output utilized in this study was SSpro (release 6.0, 2021), which provides per-residue classification into three categories: helix, strand, or other. The implementation in Python 3 was carried out following the specifications provided in its official GitHub repository [33].

## Residues exposure calculation

The solvent-accessible surface area (SASA) was calculated using the FreeSASA library, whose implementation in Python 3 was carried out following the specifications indicated in its GitHub repository [34]. Calculations were performed on all AF2 provided models of the human proteome, available on its website. For each residue, the SASA was compared to that of a completely solvent exposed residue, and classified as either exposed or buried. The classification threshold was determined from each residue SASA probability distribution as derived from analysis of thousands of proteins. Thresholds for each of the 20 residues are available in the SASA folder of the MotSASi2.0 GitHub repository.

## Gene ontology terms

Gene Ontology (GO) terms were retrieved from The Gene Ontology Resource website [35]. The file used was *go.obo* (September 2023), which was parsed in Python 3 using the *obonet* library. Terms related to cellular localization were extracted for each studied protein.

## Protein structure and matrices visualization

Protein structure visualizations and substitution matrices were generated using Visual Molecular Dynamics (VMD) software, PyMOL Molecular Graphics System and the Seaborn library [36,37]. Substitution matrices were created using Seaborn's heatmap method, with precomputed ClinVar, gnomAD, FoldX, or final numeric matrices in TSV format as input.

## SLiMs analysis and filtering pipeline

A flowchart of the pipeline is provided in the Supplementary Information. The Initial SLiM set was filtered to retain only matches containing at least one variant from ClinVar or gnomAD, forming Set 0 (S0). Additionally, motifs experimentally confirmed in the ELM database were classified as Positive Set 0 (P0), representing the true positives. Clinical significance matrices were constructed using the variant data for each protein in P0, while structural stability matrices were generated using FoldX applied to PDB or AF2 structures. These motifs were analyzed for secondary structure (Scratch), solvent accessibility (FreeSASA), conservation (Jensen-Shannon Divergence), and associated GO terms (The Gene Ontology Resource database). Candidate motifs were also evaluated through the filtering steps described below. This approach is grounded in the established characteristics of SLiMs, which are typically located in disordered, solvent-accessible regions, exhibit a sufficient degree of evolutionary conservation, and are associated with specific cellular compartments. These attributes are systematically assessed in each candidate motif and compared against the features observed in the P0 reference set.

Once the matrices were built, motifs were filtered in a three-step iterative cycle:

1. **Clinical Significance Cycle**: for all motifs in S0 their variants were classified as pathogenic or benign using clinical significance from ClinVar or gnomAD allele frequency thresholds. In this cycle, only variants located in rigid motif positions were assessed using clinical significance matrices. Motifs showing no contradictions against the true positive set (i.e., passing both the variant matrix and features evaluations) result in Positive Set 1 (P1). Motifs with discordant variant classifications were assigned to Negative Set 1 (N1), while unresolved cases were allocated to Remaining Set 1 (R1). Variants within motifs in P1 were used to refine ClinVar and gnomAD matrices. The cycle iterated until no new positive motifs could be identified.

2. **Structural Analysis Cycle**: motifs in R1 were evaluated by comparing their variants against the FoldX stability matrix obtained from the corresponding motif-receptor complex structures. As in the previous cycle, only variants in rigid positions were assessed. Tolerated SAS are those whose ΔΔG is below predefined thresholds of 2.1 kcal/mol for PDB structures and 1.6 kcal/mol for AF2 models (both calibrations are shown in Supplementary Information, Fig C in S1 Text and Fig D in S1 Text). Motifs whose variants do not contradict the resulting SAS tolerance matrix were assigned to Positive Set 2 (P2). Discordant and unresolved cases were transferred to Negative Set 2 (N2) and Remaining Set 2 (R2). As for the clinical significance cycle, motifs in P2 were used to further refine the matrices. In this cycle, only the well-defined residue positions present in the regular expression are analyzed. Flexible positions (i.e., x or ^P) in the regular expressions are analyzed in the last cycle.

3. **Flexible Position Cycle**: motifs in R2 were finally filtered analyzing the SAS (in)tolerance as evaluated using FoldX ΔΔG values in the regular expression flexible positions. The methodology mirrored that of the previous cycles, ultimately resulting in the final sets: Positive Set 3 (P3), Negative Set 3 (N3), and Remaining Set 3 (R3).

Motifs in P3 were designated as functional with high confidence, while those in N3 were classified as non-functional with equal confidence.

## Comparison against AlphaMissense: evaluation pipeline design

To perform a comparison with the bioinformatic pathogenicity predictor AlphaMissense, we developed a modified pipeline. In each cycle and iteration, a subset of 20% of ClinVar pathogenic variants and gnomAD benign variants, identified in

ELM instances (true positives) or newly detected high-confidence motifs, is set aside to form an independent evaluation set. These variants are excluded from the pipeline and subsequently used to assess the performance of the MotSASi SAS matrices in comparison with AlphaMissense predictions, utilizing metrics such as accuracy, F1-score, and Matthews Correlation Coefficient (MCC). All selected variants are located within motifs that satisfy the full set of structural, evolutionary, and annotation-based filters described in the pipeline, including conservation, secondary structure, solvent accessibility, and GO term coherence with true positive motif class instances.

## Results

### 1) Alpha fold provides high-quality SLiM-receptor complex structures

While the overall quality of AF2 models has been extensively tested, igniting a plethora of studies based on those structures, it remains essential to evaluate whether the generated predictions are suitable for the particular purposes for which they are intended [38–40]. In line with this reasoning, we first used AF2 to replicate the core motif-receptor structures of 17 ELM motif classes (the same group analyzed in our previous work, and some additional cases), each of which had at least one crystallographic structure deposited in the PDB using only their sequence information as input. Key to our approach is the accurate positioning of the SLiM within its corresponding binding site (Fig 1A, left panel). Analysis of AF2 models reveals that, while AF2 reliably predicts the overall receptor structure, improper SLiM positioning is observed in certain cases (red SLiM in the model shown in Fig 1A). This mispositioning often leads to AF2 failing to identify the correct motif binding site.

To address this issue and avoid using motif-receptor complexes with incorrectly positioned motifs, we analyzed several parameters of the AF2 models, in order to assess their ability to discriminate between correct and incorrect SLiM positioning. These parameters include the interaction energy between the motif and domain chains (computed as described in FoldX Free-Energy Change calculations in the Methods section) and the confidence of motif residues estimated by AlphaFold's pLDDT metric. As shown in Fig 1B (right panel), correct models exhibit a higher mean pLDDT for AF2-predicted SLiM residues ($82.8 \pm 10.0$ vs. $40.9 \pm 14.0$, $p < 1 \times 10^{-39}$) and a lower interaction energy between the receptor and the SLiM ($-12.1 \pm 4.1$ kcal/mol vs. $-3.1 \pm 6.6$ kcal/mol, $p < 1 \times 10^{-15}$). Based on these observations, we defined threshold values of 65

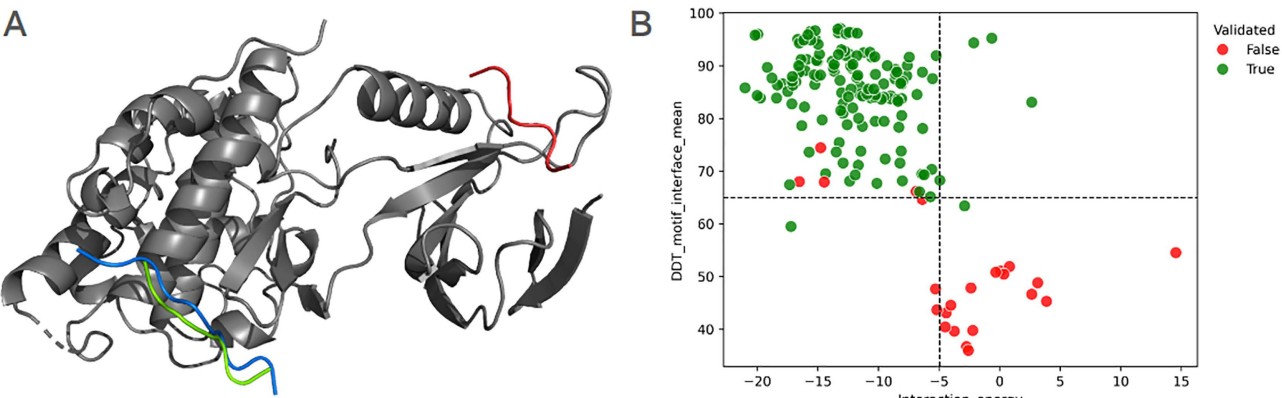

**Fig 1. (A) Crystallographic structure (pdbid 1UKH), depicting the interaction between the DOC_MAPK_JIP1_4 motif (in blue) of mouse JIP1 (UniProtID: Q9WVI9) and the Pkinase domain (in grey) of human MAPK8 (UniProtID: P45983).** Two AF2 predictions for the motif are shown: a correct prediction (in green), and an incorrect prediction (in red). (B) Scatter plot showing the relationship between the mean pLDDT of the motif residues (y-axis), and the motif-receptor interaction energy in kcal/mol (x-axis). Each dot represents an individual AF2 model: green dots indicate models manually classified as correct, while red dots denote models classified as incorrect. Dashed lines indicate threshold values selected to distinguish between correct and wrong AF2 predictions.

for the mean pLDDT of the motif residues, and -5 kcal/mol for the interaction energy. The former reflects AF2's self-assessed confidence in positioning the motif residues at that location relative to their domain binding partner, while the latter accounts for the thermodynamics contributions facilitating the complex formation. In other words, once several AF2 models are built for a given SLiM-receptor complex (we generated 10 models per complex), we selected those models that meet the mean pLDDT threshold (>65) for motif residues, and surpass the binding energy threshold (<-5 kcal/mol). Indeed, complex structures with properly positioned SLiMs could be selected using only these two parameters in all tested cases.

At the heart of MotSASi are the structural SAS tolerance matrices. We thus compared the corresponding matrices obtained from the crystallographic structures deposited on PDB with those derived from the AF2 models that passed the previous quality checks. Visual inspection of the two matrices (shown in Supplementary Information, Figure E in S1 Text) clearly reveals that they are highly similar. To quantify this similarity, we used two metrics. First, we computed the Spearman correlation coefficient between the ΔΔG values obtained with either matrix, yielding a value of 0.73 +/- 0.15. We selected the non-parametric Spearman test because, while a linear correlation holds within the -2–6 kcal/mol range, it breaks down at higher values—where the deleterious effect is unquestionable— of important note is that a strong mono-tonic relationship remains. Secondly, we classified each SAS as tolerable or non-tolerable for each matrix, defining those derived from the crystal structure as the reference (or true) cases and those from AF2 as the predicted cases. The resulting statistical performance values (e.g., an accuracy of 0.79) demonstrate that AF2 models are sufficiently accurate to be incorporated into the MotSASi pipeline.

## 2) SLiM natural variant analysis supports AlphaFold2 models

Having shown that AF2 models yield accurate SLiM-receptor structures and their resulting SAS tolerance matrices, we further analyzed whether examining known SLiM variants could aid in selecting the correct AF2 complex models. Comparing the FoldX stability matrix with the observed variant (from ClinVar and gnomAD) pathogenicity matrix provides an independent and stringent test of the AF2 model. Fig 2 presents a visual example of the matrix comparisons for correct and incorrect models.

Interestingly, although the structures and positioning of the SLiM relative to the receptor differ completely between the two models, the matrices still display some similarity. This observation arises from the inherent properties of residues and their general impact on protein structure (e.g., substituting a rigid Proline at row 2, as shown in the example of Fig 2, is typically more destabilizing than mutating the degenerate residue in row 3). Moreover, this effect is implicitly incorporated into AlphaFold2's modeling of the binding event, as it inherently considers the underlying thermodynamic principles. It further underscores the importance of carefully comparing the FoldX stability matrix with the observed variant pathogenicity matrix. This comparison is illustrated in the lower panel using a binary color scheme: green cells indicate agreement between matrix predictions, while red cells indicate disagreement. The correct model achieves 6 matches out of 8 tested variants, whereas the incorrect model achieves only 3. Similar trends are observed for other motifs. Unfortunately, as previously discussed, observed variant pathogenicity matrices tend to be scarce (i.e., they include data for only a limited subset of possible variants), making them unsuitable for individual assessments. As a result, we performed a broader evaluation by aggregating all observed variants within the motif classes considered for the validation process. We calculated a global accuracy metric for those AF2 models that passed, and those that did not pass the confidence and inter-action energy thresholds (as established in the previous section). AF2 models that met both criteria achieved an overall accuracy of 77.2%, while models that did not meet the thresholds achieved an overall accuracy of 58.2%. This difference was statistically significant, as shown by the proportions comparison test (p-value < 0.0005).

Summarizing both analyses, it is clear that, taken together, the motif residues' mean pLDDT, the SLiM-receptor interaction energy, and the comparison of the FoldX and clinically significant variant matrices, allow for the selection of the best AF2 models.

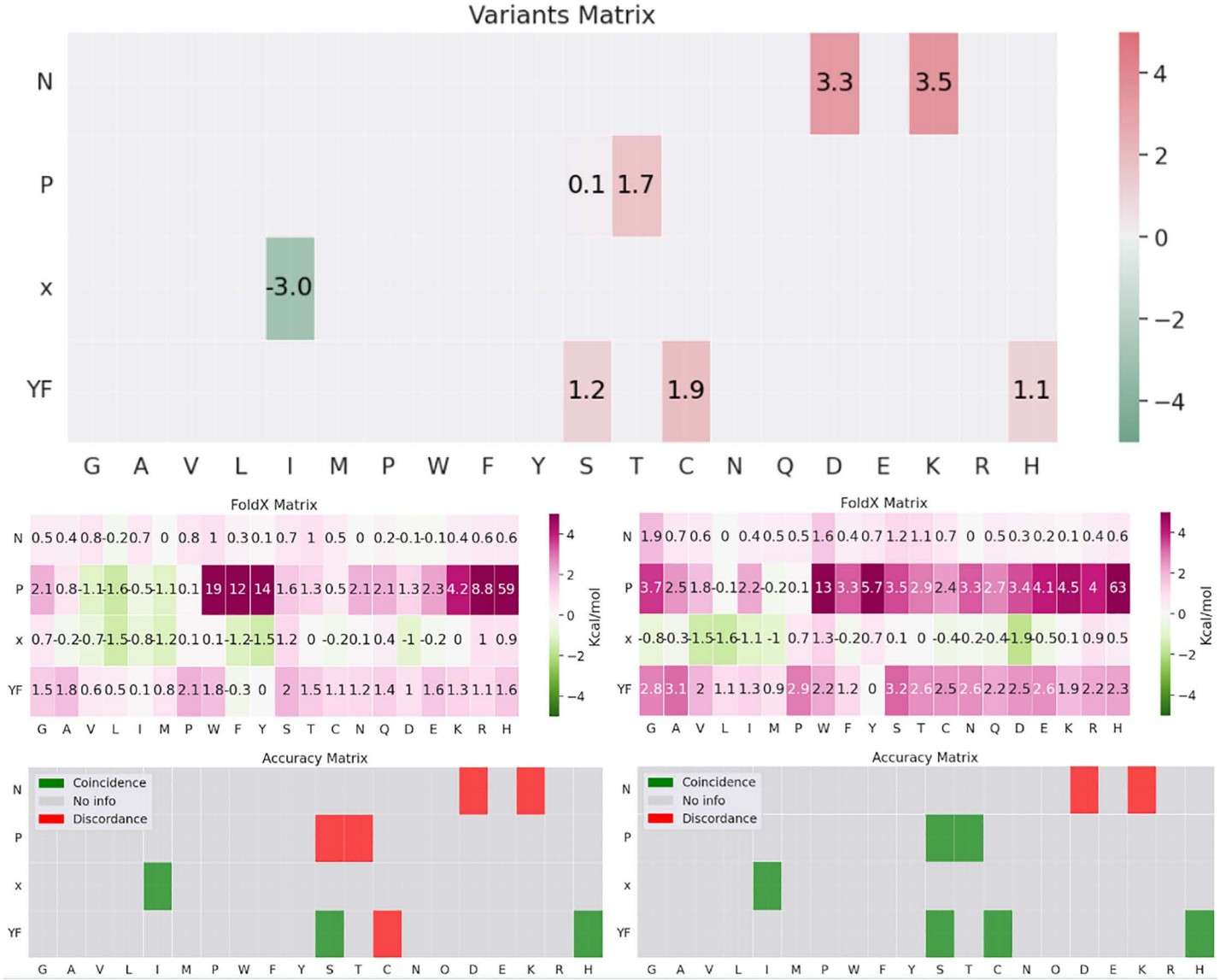

**Fig 2. Comparison of the observed variant pathogenicity matrix (upper panel, expressed in ClinVar confidence units, as described in the Methods section), with the FoldX stability matrices (middle) for the correct (left panel) and incorrect AF2 models (right panel), with stability values expressed in kcal/mol for the LIG_PTB_Apo_2 motif class (regular expression: NP.[YF]).** The lower panel summarizes the coincidences and inconsistencies between observed ClinVar and gnomAD variants and FoldX matrices.

### 3) AlphaFold2 allows increased high-confidence SLiMs prediction

One of the main limitations of the previous version of MotSASi was the scarcity of motif-domain structures available in the PDB. This limitation constrained both the computation of FoldX stability matrices, and the identification of new high-confidence candidates. The previous version of MotSASi included 20 different SLiM-receptor complexes, encompassing 4,389 analyzed motifs hosted by 3,085 unique proteins. By updating the databases and incorporating AF2-generated models of motif-receptor complexes, we have now analyzed 51 distinct ELM motif classes—27 with crystallographic structures deposited in the PDB and 24 based on AF2-generated models—resulting in a total of 8,731 analyzed motifs

corresponding to 5,027 unique proteins (Table 1). Considering that the current human proteome in SwissProt comprises 20,435 entries, our predictions cover approximately 25% of the human proteome. Furthermore, among the 4,866 gene-encoded proteins associated with Mendelian diseases reported in OMIM, 1,694 contain at least one high-confidence motif, increasing the coverage to 35% of disease-associated proteins. This information is summarized in Table 1.

These results, together with the statistics presented in Fig G in S1 Text, allow for a more comprehensive analysis of the features that characterize accurately predicted, positively selected SLiMs, as well as those that are discarded during the filtering process. Starting from the total number of Regular Expression (RE) matches (considered as 100% of the potential SLiMs), a large fraction (ca. 80%) cannot be further analyzed due to the absence of clinically relevant variants. Among the remaining SLiMs, another ca. 80% are ultimately discarded—an expected outcome given the low specificity of REs. The primary reason is insufficient evolutionary conservation (83% of evaluated candidates), followed by discordant GO term annotations (71%), incompatible secondary structure (69%) relative to known instances of the motif class. Finally, a small fraction is discarded due to limited solvent and thus receptor accessibility (23%). Notably, particular SLiMs candidates are usually excluded based on more than one of these criteria (e.g., being both poorly conserved and structurally buried). Conversely, the high-confidence motifs retained by MotSASi—representing approximately 2% of the total initial candidates—are those that exhibit strong coherence with the features observed in true positive motif instances. These include baseline levels of conservation, structurally compatible secondary structures (typically disordered), solvent accessibility, and consistent subcellular localization based on GO annotations.

The MotSASi pipeline presented herein, with the key inclusion of AF2 models, expands the coverage of human ELM instances nearly 22-fold, with enrichment levels ranging from 19-fold for the PDB group to 29-fold for the AF2 group. Encouragingly, the pipeline discards approximately 14% of regular expression matches as potential false positives, deeming them unlikely to represent functional SLiMs. Nonetheless, significant limitations persist, as approximately 84% (315,957) of the SLiM regular expression matches processed by MotSASi lack known variants and, as such, cannot yet be fully analyzed. Despite these challenges, the increased coverage of high-confidence potential SLiMs and the identification and elimination of false positives are expected to improve further as more variants are discovered and classified. Nevertheless, as a result of our work, we now have access to a large number of newly identified motif instances or SLiMs,

**Table 1. Summary of the analyzed ELM motif classes, instances, and associated human proteins.**

|  | MotSASi | PDB | AF2 | OMIM |
|---|---|---|---|---|
| # ELM classes | 51 | 27 | 24 | 42 |
| # ELM instances | 405 | 305 | 100 | 199 |
| # RE proteome hits | 377,170 | 165,481 | 211,689 | 124,171 |
| # No variants motif instances (% of total) | 315,957 (84%) | 137,444 (83%) | 178,513 (84%) | 98,554 (79%) |
| # Negative motif instances (N3) (% of total) | 52,482 (14%) | 22,218 (13%) | 30,264 (14%) | 22,047 (18%) |
| # Positive motif instances (P3) (% of total) | 8,731 (2%) | 5,819 (4%) | 2,912 (1%) | 3,570 (3%) |
| # Proteins | 5,027 | 3,896 | 2,360 | 1,694 |
| Enrichment against ELM inst. | X 21.56 | X 19.08 | X 29.12 | X 17.94 |

Table caption: number of motif types (# ELM classes), experimentally validated SLiMs (# ELM instances), regular expression matches in the SwissProt human proteome (# RE proteome hits), and their categorization: matches without ClinVar/gnomAD variants (# No variants motif instances), false positives (# Negative motif instances), and high-confidence SLiMs (# Positive motif instances). The "Proteins" row shows the total number of SwissProt human proteins harboring at least one motif per category. Columns specify the datasets analyzed: MotSASi (entire dataset), PDB (motifs with crystallographic structures), AF2 (motifs modeled with AlphaFold2), and OMIM (motifs in proteins linked to diseases in OMIM).

many of which are located in disease-associated OMIM genes, underscoring their potential functional significance. Their identification provides, therefore, a working hypothesis for further studies, seeking to confirm this function in the proper cellular and physiological context, while also providing a more detailed protein`s functional annotation (see below for the results concerning two particular examples).

## Helical calmodulin-binding motif

As an example of a newly analyzed ELM class using AF2, we present the LIG_CaM_NSCaTE_8 motif (ELM RegEx: W[^P][^P][^P][IL][^P][AGS][AT]). Interestingly, two experimentally validated human instances of this motif were previously reported in the ELM database, located in the alpha subunit of L-type voltage-gated calcium channels (VGCCs). One instance is found in the N-terminal cytoplasmic domain of Cav1.3 (HGNC gene name: *CACNA1D*, UniProtID: Q01668) and is known as the N-terminal spatial Ca²⁺-transforming element (NSCaTE) motif. This motif has been extensively studied and shown to play a key role in accelerating channel closure when the N-lobe of Ca²⁺-CaM binds to the NSCaTE motif, a regulatory mechanism called calcium-dependent inactivation [41]. Loss-of-function variants in *CACNA1D* are responsible for a Mendelian disorder known as SANDD (SinoAtrial Node Dysfunction and Deafness, MIM Number: 614896), which follows an autosomal recessive inheritance pattern [42]. Notably, *CACNA1D* is expressed in neuronal cells and cardiac myocytes.

Through our iterative analysis, we were able to discard 74 motifs and identify 7 new high-confidence potential candidates for this SLiM in the human proteome. For instance, Ca²⁺-CaM, a ubiquitously expressed protein critical for various cellular functions, interacts with numerous targets. These targets generally consist of sequences of 15–30 amino acids with an intrinsic propensity to form an alpha helix. Our AF2 model and the accompanying amino acid substitution matrix provide a deeper understanding of the motif-domain binding event. As shown in Fig 3, Trp53 occupies a central position within the hydrophobic pocket formed by the EF-hand of CaM. Regarding the substitution matrix, we observe that Trp in the first position, Ile or Leu in the fifth position, and small hydrophobic amino acids in the eighth position are highly

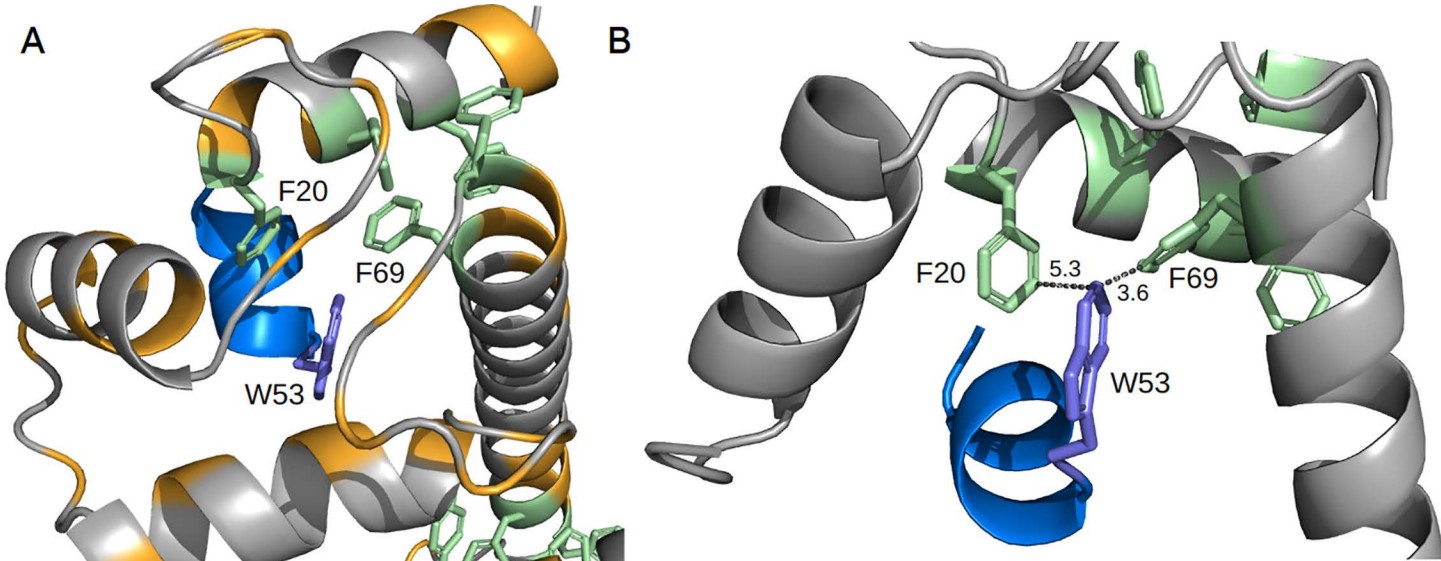

**Fig 3.  (A) Trp53 (shown in purple) of Cav1.3 (UniProt ID: Q01668) occupies a central position within the hydrophobic pocket formed by the EF-hand of CaM (UniProt ID: P0DP25).** The motif chain is shown in blue, the domain chain in grey, aromatic residues of the domain in green, and aliphatic residues of the domain in orange. (B) Trp53 viewed from another angle, with distances to neighboring aromatic residues depicted (measured in Å).

constrained. These positions exhibit limited allowance for alternative amino acids. In contrast, the fourth position, previously defined as highly degenerate by ELM, appears less permissive, favoring only small hydrophobic residues. Based on our SAS matrix analysis (Figure F in S1 Text), we propose the following refined regular expression for the NSCaTE motif: W..[APC][VLIM][^P][^P][GAVMSTC].

## 4) SLiM-receptor complex models improve SLiM variant pathogenicity prediction

MotSASi, as an algorithm, leverages clinical significance and population allele frequency data of variants reported in relevant databases. Nevertheless, integrating crystallographic structures to facilitate thermodynamic predictions of variant deleteriousness remains critical. In this context, we sought to compare MotSASi with another method that relies on structural information to train its algorithm. Given its recent success, we selected AlphaMissense (AM) as the candidate, although other bioinformatic prediction tools recommended by ClinGen for assessing clinically relevant variants—such as REVEL, BayesDel, MutPred2 and VEST4—could also have been used [43]. To ensure a fair comparison, we randomly set aside a subset comprising 20% of ClinVar pathogenic and gnomAD benign variants found in ELM instances (true positives), and/or newly identified high-confidence motifs (see Methods). The final test set consisted of 2,335 variants, including 2,155 gnomAD benign and 180 ClinVar pathogenic variants. MotSASi outperformed AlphaMissense across all three evaluated metrics: accuracy (97% vs. 76%), F1-score (0.98 vs. 0.83), and Matthews Correlation Coefficient (MCC) (0.78 vs. 0.39). It is clear that while AM is an excellent general-purpose variant pathogenicity prediction tool, the detailed analysis of underlying structures in a biological context, as performed by MotSASi, significantly improves overall prediction accuracy Table 2).

Analysis of the confusion matrix shows that the key to the observed improvement in MotSASi lies in its ability to avoid tagging known benign variants as pathogenic. The naturally imbalanced dataset, with many more benign variants as often observed in real-world examples, results in AM misclassifying 285 benign variants as pathogenic, whereas MotSASi shows only 6 cases of these potential false positives. As expected, for known pathogenic variants, while MotSASi fails to detect 60, AM only fails in 26 cases. Generally speaking, it appears that MotSASi tends to be more conservative than AM concerning pathogenic classification, which aligns with current ClinGen recommendations. The overall data show that with MotSASi, we have developed an ideal tool for studying missense variants residing in functional SLiMs, even when a crystallographic structure has not been deposited in the PDB, provided we can generate an optimal model with AF2.

To gain deeper insight into the comparison, we examined specific cases. A well-studied example is the LIG_PTB_Apo_2 motif in LDLR, which is crucial in its internalization upon ligand binding. The motif follows the regular expression NP.[YF], and the MotSASi-defined SAS tolerance matrix allows refining it to N[PA][^GWFYKH][YFWML]. Several variants affecting this motif have been previously reported in ClinVar and gnomAD. At the first position, Asn825Asp and Asn825Lys variants have been reported as pathogenic. The substitution to lysine is correctly predicted by AM as Likely pathogenic, whereas aspartate is incorrectly classified as Likely benign. At the second motif position, Pro826Ser and Pro826Thr have been reported as pathogenic, and again the first variant is correctly predicted as Likely pathogenic by AM, while the latter is classified as Uncertain significance. The third motif position lacks clinically significant variants; however, a Leu827Ile

**Table 2. Confusion matrices comparing the performance of MotSASi and AlphaMissense on a validation set of variants.**

|  | Pathogenic MotSASi verdict | Uncertain MotSASi verdict | Benign MotSASi verdict | Pathogenic AM verdict | Uncertain AM Verdict | Benign AM verdict |
|---|---|---|---|---|---|---|
| ClinVar Pathogenic | 120 | 34 | 26 | 154 | 12 | 14 |
| gnomAD Benign | 6 | 6 | 2143 | 285 | 242 | 1628 |

variant with a high allele frequency is reported in gnomAD. It is correctly classified as Likely benign by AM. At the fourth and final motif position, three missense variants (Tyr828Ser, Tyr828Cys, and Tyr828His) have been reported. All three are classified as Uncertain by AM. In contrast, MotSASi correctly predicts all these variants by leveraging the information of the SAS tolerance matrix.

This misclassification could stem from the fact that AM bases its predictions on individual protein structure assessments without explicitly considering protein-protein binding events. Inspection of the 1NTV crystallographic structure deposited in the PDB, which represents the corresponding motif-domain interaction, can help us better understand this trend. For the misclassified Asn825Asp variant, we observe that the mentioned Asn is involved in a double hydrogen bond. While its oxygen donates an electron pair to the backbone nitrogen of the Val at the third position, it simultaneously acts as a hydrogen donor through its amide nitrogen with a backbone oxygen from the domain chain. Additionally, there is a Phe residue in close proximity, which most likely is stabilized in the neutral environment. Therefore, it seems highly unlikely that a negatively charged residue, such as Asp, could be tolerated, leading to the observed pathogenic variant. Regarding the second motif position, it appears that the proline fulfills a specific stereochemical role, due to its particular backbone conformation, enabling the association between the first and third positions while occupying minimal space. It seems reasonable to assume that only Pro would be accepted, even though MotSASi's energetic considerations suggest that other small aliphatic residue might also be possible. At the third position, occupied by a Val, we find it in close proximity to hydrophobic residues. This suggests that not all residues would be allowed, but rather only aliphatic ones of a specific size. AM fails to recognize this and cannot predict the pathogenicity of either Thr or Ser substitutions. Finally, at the fourth position, the Tyr residue appears to occupy a pocket adapted for an aromatic residue, adding an hydrogen bond between the Tyr hydroxyl group and the carbonyl oxygen in the domain backbone. As reflected by the MotSASI SAS tolerance matrix it seems reasonable to conclude that mutations to non-aromatic residues would likely disrupt this interaction.

## Discussion

In this work, we present our progress in studying SLiMs in human proteins, leveraging increasingly enriched variant databases, and the recently developed Artificial intelligence (AI) algorithms for protein structure prediction. Our focus, oriented towards the broad spectrum of Mendelian disorders, specifically addresses missense variants located in SLiMs. These functional elements, which play key roles in numerous physiological processes, are vulnerable to disruption by genetic alterations. When these changes occur in genes associated with a given disease, they can lead to the development of the underlying pathology, making them clinically relevant.

Our pipeline significantly expanded the repertoire of predicted functionally relevant SLiMs, achieving an almost 22-fold increase in the number of candidate motifs analyzed across the whole human proteome. These predictions remain to be validated by functional assays to confirm their physiological relevance. While this progress is still far from capturing the entirety of potential functional SLiMs—predicted to be around one million—it represents an important step in providing experimental scientists with hypotheses and advancing our understanding of protein-protein interactions. As highlighted in our previous work, a key strength of the MotSASi method lies in its ability to integrate reported missense variants and three-dimensional protein structural data to generate more accurate predictions. Conversely, it is crucial to emphasize the value of filtering out non-functional motifs, as this prevents researchers from expending unnecessary effort on dubious Variants of Uncertain Significance (VUS). Given the clinical focus of MotSASi, we restricted our analysis to motif matches associated with clinically interpretable variants. Our method showed comparable performance when using both PDB and AF2 structural inputs.

Generating AF2 models introduces another layer of complexity. Although ELM provides curated data, gaps often require manual intervention. For example, missing interaction pairs between motifs and binding domains, outdated transcript references, and alternative (no-Pfam) domain annotations complicate the preparation of FASTA files for AF2 predictions. In some cases, generating an AF2 prediction with the desired minimum quality levels of motif-domain binding

energy and pLDDT confidence values for motif residues is impossible. Biological challenges also arise, such as motifs requiring post-translational modifications (e.g., phosphorylation), which AF2 cannot currently model. Attempts to use phosphomimetic residues to approximate phosphorylated states in PDB structures yielded poor results when phosphates were critical for binding. Similarly, motifs involving non-standard stoichiometries, such as interactions with dimeric domains, posed difficulties. Nevertheless, we successfully addressed these cases by modeling sequences with domain repeats, demonstrating AF2's utility when provided with accurate input.

It is also important to mention that during the writing process of this work, AlphaFold3 (AF3) appeared in the public domain. Although initially (May 2024), it could only be used for limited online tasks, the full set of parameters was released in November 2024. AF3 is expected to outperform its predecessor in multimeric modeling and provide more accurate peptide-protein binding site predictions, potentially enhancing MotSASi's accuracy. On the other hand, incorporating AF3 into MotSASi's pipeline would serve as an additional test for its structural prediction capabilities, particularly in the context of comparing the structural and clinical substitution matrices. Moreover, it has been documented that AF2 still struggles to accurately predict the conformation of intrinsically disordered protein regions, where, as previously mentioned, several SLiMs occur [44]. These cases might also benefit from AF3. Finally, it is worth mentioning that although MotSASi's pipeline is currently based on AF2, alternative structure prediction tools such as RoseTTAFold or ESMFold among others, could also be used.

The high-confidence SLiM candidates and missense substitution matrices provided by MotSASi contribute to the clinical genomics community with tools to refine ACMG guidelines for variant interpretation [45]. These widely adopted standards in clinical molecular diagnostics emphasize that the interpretation of clinically relevant variants must be conducted according to a set of criteria, including familial segregation, variant allele frequency, molecular effect, and others, which are implemented in the form of 28 different alphanumeric labels. These criteria, or labels, are commonly referred to as the ACMG labels/criteria for variant annotation. Among these, one particularly relevant to MotSASi is the pathogenic and moderate PM1 label. PM1 label is assigned for those variants "located in a mutational hot spot and/or critical and well-established functional domain without benign variation". This label has not been reviewed by ClinGen experts in recent years, which is a gap of concern for our group. Traditionally, PM1 candidate residues or regions were assigned based on known enzyme active sites, recurrent mutational "hot spots" or other protein residues/regions with a well-established functionality usually including single mutagenesis experiments. SLiMs are good candidates as small functional domains, i.e., PM1 candidates. However, even though it may be tempting to consider the whole motif for PM1, our results clearly show that further refinement is needed, and possible as shown by MotSASi. Our assignment of high-confidence motifs combined with the final SAS tolerance matrices allows us for each human protein with high confidence SLiM to determine which residues (and which SAS) are non-tolerant, and thus their underlying variants should be labeled with PM1. In this context, and to help clinical genomic analysts implement this labeling criteria, we provide a file (as part of the SI) with all predicted variant effects in high-confidence SLiM candidates and known instances cataloged in the ELM database. In other words, in a clinical setting, we propose assigning PM1 to variants located within high-confidence functional motifs and a high non-tolerant confidence score value in the final SAS matrix.

Another interesting avenue for future research regarding variant pathogenicity assessment involves the receptor (or domain) side of the SLiM–receptor interaction. Variants occurring on the receptor side may also disrupt SLiM binding and contribute to disease. Notably, if a variant negatively affects binding at the receptor level, it could impair interactions with all SLiMs targeting that receptor—potentially altering the entire interaction network. This may lead to a distinct phenotype (and disease manifestation) compared to variants located within individual SLiMs.

Finally, given our strong emphasis on the structural component of motif-domain interactions, we compared MotSASi with AM. Previous studies evaluating AM indicate that the algorithm performs less accurately in disordered regions, as defined by AF2 using residue confidence scores below a specific threshold [46]. This reduced performance is likely due to the enrichment of benign variation in disordered proteome regions, which paradoxically also harbor functional sequences

such as SLiMs, where pathogenic variants can disrupt motif-domain interactions. Although MotSASi demonstrated markedly higher accuracy than AM (97% vs. 76%), this result may be misleading due to the high enrichment of benign variants in the test set. The literature has extensively discussed this issue, particularly regarding the typical imbalance between positive and negative instances in clinical genomics datasets [47]. In such cases, alternative metrics are recommended, such as the Matthews Correlation Coefficient (MCC), which rewards methods that correctly predict both positive and negative instances. Using this metric, MotSASi also outperformed AM (0.78 vs. 0.39).

Analysis of the confusion matrices generated by each method revealed that, while AM provides more accurate predictions for pathogenic variants in the test set, it also produces many false positives. It is important to emphasize that in clinical genomics, both types of misclassifications are problematic, but in different contexts. In the case of rare diseases diagnostics, where usually a clinical diagnosis is already available, false negatives are generally preferable to false positives and it is more prudent to withhold reporting a candidate variant until stronger evidence is available, rather than reporting it prematurely and risking subsequent retraction. However, false negatives also carry clinical risk, particularly when they lead to missed opportunities for timely intervention. This is especially critical in the context of actionable genetic findings—such as pathogenic variants in BRCA1 or BRCA2, which are included in the ACMG list of secondary findings—where early detection enables established preventive strategies aimed at significantly reducing morbidity and mortality [48].

Nonetheless, the observed performance difference between AM and MotSASI, is also an interesting point of concern. A deeper analysis of the accuracy differences between the two methods reveals two main contributing factors. First, upon further examination of the benign variants misclassified by AM, we found that over half of them are labeled as of Uncertain Significance (i.e., VUS). This suggests that part of the performance discrepancy arises from the distinct thresholds each method uses to classify a variant as benign. AM applies a general threshold designed to operate across the entire human proteome, regardless of protein region or functional context. In contrast, MotSASi uses a threshold specifically optimized for SLiMs in the context of their interactions with protein partners. The second key factor relates to the intrinsic nature of SLiMs and their interactions with their receptor domains, as reflected in the RE that define them. Typically, some positions within a RE are structurally constrained and tolerate only a limited set of amino acids ("rigid positions"), whereas others are more permissive, allowing for greater sequence variability ("flexible positions"). We assessed the performance of MotSASi and AM separately on variants located in each type of position. While both algorithms performed similarly on flexible positions (MCC: MotSASi 0.50 vs. AM 0.37), the difference was considerably more pronounced for rigid positions (MCC: MotSASi 0.88 vs. AM 0.41). These results are consistent with the design of each method. AM is a general-purpose predictor that evaluates the variant in isolation, without explicitly modeling protein–protein interactions. By contrast, MotSASi directly assesses the variant within the structural context of the motif–domain complex, enabling it to better capture the functional consequences of mutations at key binding residues. Since flexible positions typically do not make direct contacts with the receptor, both methods show comparable predictive performance for those variants.

Last but not least, it is important to acknowledge that the dataset used to compare both methods may be inherently biased in favor of MotSASi, as all tested variants were selected from motif matches that passed conservation, structural, and annotation filters. These filters align with typical SLiM characteristics—such as their location in disordered or loop regions—where AM has been previously reported to show lower performance. Given that AM has been trained on a vast dataset of missense variants, it is likely that its performance would improve if trained exclusively on SLiM-residing missense variants. However, it is essential to consider the broader context: while MotSASi is a more suitable tool for assessing variants located within SLiMs, due to its specific modeling of the underlying biological hypothesis, AM remains an excellent bioinformatics prediction tool with a considerably broader scope.

In summary, after the first MotSASi publication, we predicted that advancements in massive sequencing technologies, their integration into clinical practice, increasing experimental validation of predicted SLiMs, and new motif-domain crystal structures would collectively enhance our understanding of amino acid substitution effects within motifs. This, in turn, would lead to more precise and accurate predictions of biologically relevant SLiMs. Remarkably, most of these

developments have materialized in just three years—a surprisingly short time frame. Envisioning the future possibilities and direction of cutting-edge SLiM analysis remains challenging. Will refined AF2-like predictions for individual motif-domain pairs take center stage? Could multiplex assays of variant effects (MAVEs) be applied to SLiM studies, contributing directly to functional assay evidence for clinically relevant variants? Will large-scale functional assays identify most functional motifs across the human proteome? Until that time comes, significant efforts will need to be sustained, and MotSASi will play a pivotal role in striving to achieve the highest possible diagnostic yields in clinical genomics.

## Supporting information

**S1 Text.** Figure A SI. Flowchart of the MotSASi pipeline, integrating crystallographic structures from the PDB and predictive models ftrom AlphaFold2. CSC = Clinical Significance Cycle (x = 1), SAC = Structural Analysis Cycle (x = 2), FPC = Flexible Position Cycle (x = 3), JSD = Jensen-Shannon Divergence, P = Positive set, N = Negative set, R = Remaining set. The CSC, SAC, and FPC cycles occur sequentially. Candidates in the S0 and Rx sets are assessed using ClinVar and gnomAD matrices in all three cycles, whereas candidates in the R1 and R2 sets are evaluated against the FoldX matrix exclusively in the SAC and FPC cycles. **Figure B SI.** Plots of functions and their corresponding formulas for (A) gnomAD, (B) PDB-derived, and (C) AlphaFold2 (AF2)-derived Confidence Scores. **Figure C SI.** (a) Histogram displaying the ClinVar and gnomAD variants used to define the ΔΔG stability threshold (in kcal/mol). Pathogenic variants are indicated in red, and benign variants in green. (b) ROC curve (AUC = 0.929) generated during the ΔΔG threshold determination process for PDB crystallographic structures. The red dot marks the threshold value of 2.1 kcal/mol, corresponding to a sensitivity of 83% and a specificity of 84%. **Figure D SI.** (a) Histogram of ΔΔG stability values (in kcal/mol) calculated on Alpha-Fold2 (AF2)-generated models for the same ClinVar and gnomAD variants previously classified based on the 2.1 kcal/mol PDB threshold. Non-tolerated variants are shown in red, tolerated variants in green. (b) ROC curve (AUC = 0.865) generated during the process of determining the ΔΔG cutoff value for AF2. The red dot marks the cutoff value of 1.6 kcal/mol, corresponding to a sensitivity of 81% and a specificity of 80%. **Figure E SI.** Comparison of structural (stability ΔΔG) SAS matrices obtained using either the crystallographic structure deposited in the Protein Data Bank (PDB) (upper panel) or the SLiM-receptor structure modeled by AlphaFold2 (AF2) (lower panel) for the LIG_PDZ_Class_1 ELM motif class (regular expression: [ST].[ACVILF]$). **Figure F SI.** Final SAS matrix with confidence scores in the cells for the LIG_CaM_NSCaTE_8 motif class (regular expression: W[^P][^P][^P][IL][^P][AGS][AT]). **Figure G SI.** Pipeline illustrating the filtering process applied by MotSASi. Values represent the aggregate data across all motif classes and instances evaluated. (DOCX)

## Acknowledgements

To Mariano Martin, who contributed significantly to the development of MotSASi in its early stages, both in its theoretical biological foundation and its implementation at the programming level.

## Author contributions

**Conceptualization:** Franco Gino Brunello, Lorenzo Erra, Juan Nicola, Marcelo Adrián Martí.

**Data curation:** Franco Gino Brunello, Lorenzo Erra.

**Formal analysis:** Franco Gino Brunello, Lorenzo Erra, Juan Nicola, Marcelo Adrián Martí.

**Investigation:** Franco Gino Brunello, Lorenzo Erra, Juan Nicola, Marcelo Adrián Martí.

**Methodology:** Franco Gino Brunello, Lorenzo Erra, Juan Nicola, Marcelo Adrián Martí.

**Project administration:** Franco Gino Brunello, Marcelo Adrián Martí.

**Resources:** Lorenzo Erra, Marcelo Adrián Martí.

**Software:** Franco Gino Brunello, Lorenzo Erra, Marcelo Adrián Martí.

**Supervision:** Franco Gino Brunello, Juan Nicola, Marcelo Adrián Martí.

**Validation:** Franco Gino Brunello, Lorenzo Erra, Marcelo Adrián Martí.

**Visualization:** Franco Gino Brunello, Lorenzo Erra.

**Writing – original draft:** Franco Gino Brunello, Lorenzo Erra, Marcelo Adrián Martí.

**Writing – review & editing:** Franco Gino Brunello, Lorenzo Erra, Marcelo Adrián Martí.

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
