## [Decision Letter · Decision Letter 0]

2 Mar 2025

PCOMPBIOL-D-25-00157

Integrating AlphaFold2 Models and Clinical Data to Improve the Assessment of Short Linear Motifs (SLiMs) and Their Variants’ Pathogenicity

PLOS Computational Biology

Dear Dr. Brunello,

Thank you for submitting your manuscript to PLOS Computational Biology. After careful consideration, we feel that it has merit but does not fully meet PLOS Computational Biology's publication criteria as it currently stands. Therefore, we invite you to submit a revised version of the manuscript that addresses the points raised during the review process.

Please submit your revised manuscript within 60 days (May 01, 2025). If you will need more time than this to complete your revisions, please reply to this message or contact the journal office at ploscompbiol@plos.org. Please include the following items when submitting your revised manuscript:

We look forward to receiving your revised manuscript.

Kind regards,

Mohammad Sadegh Taghizadeh, Ph.D.

Academic Editor

PLOS Computational Biology

Ilya Ioshikhes

Section Editor

PLOS Computational Biology

**Journal Requirements:**

At this stage, the following Authors/Authors require contributions: Franco Gino Brunello. Please ensure that the full contributions of each author are acknowledged in the "Add/Edit/Remove Authors" section of our submission form.

4) Please ensure that all Table files have corresponding citations and legends within the manuscript. Currently, Table 2 in your submission file inventory does not have an in-text citation. Please include the in-text citation of the table.

5) We have noticed that you have uploaded Supporting Information files, but you have not included a list of legends. Please add a full list of legends for your Supporting Information files after the references list.

Potential Copyright Issues:

i) The following Figure contains a logo or branding: Supplementary Figure 1. We are not permitted to publish this under our CC-BY 4.0 license, even with permission. We ask that you please remove or replace it.

7) Thank you for stating that "The data supporting the findings of this study are available from the following sources: ClinVar (https://www.ncbi.nlm.nih.gov/clinvar/), gnomAD (https://gnomad.broadinstitute.org/), the Protein Data Bank (https://www.rcsb.org/), UniProt (https://www.uniprot.org/), the Eukaryotic Linear Motif Resource (http://elm.eu.org/), and AlphaFold (https://alphafold.ebi.ac.uk/)." Please note that your Data Availability Statement is currently missing the DOI/accession number of each dataset OR a direct link to access each dataset. If your manuscript is accepted for publication, you will be asked to provide these details on a very short timeline. We therefore suggest that you provide this information now, though we will not hold up the peer review process if you are unable.

8) Please amend your detailed Financial Disclosure statement. This is published with the article. It must therefore be completed in full sentences and contain the exact wording you wish to be published.

9) Your current Financial Disclosure states, "This work was supported by the following grants: Genómica Clínica de Enfermedades Poco Frecuentes (GeC-EpoF), Proyectos de Redes Federales de Alto Impacto, CONVE 2023-100390147-APN-MCT. F.G.B and L.E were supported by a PhD fellowship from the Consejo Nacional de Investigaciones Científicas y Técnicas (CONICET) of Argentina. M.M was supported by a Postdoctoral fellowship from . J.P.N and M.A.M are investigators of CONICET and Universidad Nacional de Córdoba and Universidad de Buenos Aires, respectively. "

However, your funding information on the submission form indicates receiving no fund. Please ensure that the funders and grant numbers match between the Financial Disclosure field and the Funding Information tab in your submission form. Note that the funders must be provided in the same order in both places as well.

**Reviewers' comments:**

Reviewer's Responses to Questions

**Comments to the Authors:**

**Please note that one of the reviews is uploaded as an attachment.**

Reviewer #1: review attached.

Reviewer #2: SLiMs are important and understanding consequences of mutations and how they affect cellular wiring may be informative for molecular disease underpinnings and development of treatment. Molecular and especially structural studies have lagged behind as these modules can be difficult to capture, in part due to their intrinsically disordered nature. The manuscript is timely as novel methodologies like AF2 enable modelling of these interactions, and it is crucial to find out how well predictions of specificity on such models work. The manuscript is well-written, and easy to follow.

The main limitation is that the work only compares predictions on experimental structures vs. AF2 models, but not to experimental data such as sequences and affinities from phage display. The consensus patterns are by necessity limited in accuracy, and clinical data, while very interesting for applications, is not systematically available. Further, it appears that a large fraction of the contribution relates to the peptide alone, rather than the interaction (see below for details). While the peptide itself will of course have important contributions, this might be misleading and also not the intention.

Major comments:

P8 “For genes with at least 10 pathogenic variants, the allele frequency threshold was determined based on the most frequent pathogenic variant reported in ClinVar.” - given how many random events affect allele frequency, such as population bottlenecks, this seems like a risky strategy. Is there a way to test how reliable this is? Are false positives or false negatives more problematic here? Would it be possible to consider sequences from great apes as controls?

Figure 2: what are the units here?

It is interesting that even the incorrect models achieve fairly high accuracy. What can we learn from this? To be the devil’s advocate, would it be enough to study the peptide in isolation? In other words, are most of the effects you find those on the monomer conformation? Of course these also affect whether there is an interaction, but it would be very nice if your work could also assess how much the domain side actually contributes? Is this largely a limitation of the tool used (FoldX), and have you explored alternatives?

Within FoldX, why do you use PositionScan and not AnalyseComplex (https://foldxsuite.crg.eu/command/AnalyseComplex) which is for interface analysis?

In this context, it is worth noting that the label “interaction energy”is misleading.

https://doi.org/10.1002/anie.202213362 looks relevant in the context of this work and should perhaps be included.

Generally, the figure legends need more detail, including for supplementary figures. For example, for Suppl. Figure S1, what are RC, FxC, FIC? In Suppl. Figure S4, where exactly is the source for “previously classified” cases? Reference?

p9 describes how ClinVar evidence is “scored”. A common minimum standard is “at least one star”. While in this work, the stars are counted, it is unclear what would happen for a variant with 5 pubs saying patho and 5 saying benign, this might still add up to 1?

Did you consider other predictors beyond AlphaMissense, e.g. ESM1v?

Minor comments

P4, “on the contrary” - flow is strange, rephrase

P13 “kery to our approach” - typo, should be key?

P15 “classified as reproducible” what does this mean? Technically reproducible? Or finding the same conformation as in the xtal structure? Yet a third option?

P17 “Summarizying” typo

P12 + Fig. S1 - Do the clinical significance cycle and structural analysis cycle happen simultaneously (Fig. S1) or one after another (text on page 12)?

Fig. S5 - again, no details. What is this motif shown as an example? I don’t think it’s in the main text, either

P15 - If I am reading this correctly, they get the Perason’s R of 0.65 with the error of 0.22 for the FoldX predictions on crystals and on AF2. This is quite low? Is this because some AF2 predictions were way off? + As a figure for this paragraph we get Fig. S5, which is one individual case out of many, and we are supposed to inspect it visually.

Fig. 2 has a bad resolution (difficult to read)

Reviewer #3: In “Integrating AlphaFold2 Models and Clinical Data to Improve the Assessment of Short

Linear Motifs (SLiMs) and Their Variants’ Pathogenicity” the authors present an improved prediction method for SLIMs with Mendelian variants. The project holds promise but the manuscript lacks important details, especially in the method section. This makes it difficult to fully evaluate the work and review the manuscript. The authors are encouraged to consider the following examples and apply them throughout their manuscript:

1. Some things are described in methods but not presented in results. For example:

a. For the SLIM predictions described in methods, what was found? This could at least be presented in the supplementary materials.

b. For some SLIMs, experimental models were found and used while for others AlphaFold models were used, how many of each? This could at least be in the supplementary materials.

2. Methods are not sufficient. For example:

a. The AlphaFold2 (AF2) models: it is mentioned that the best complex is selected and the mean plddt for best complexes is 65, which is in the low confidence range. What are the confidence levels of the SLIMs and their models – this should be included in the manuscript.

b. Further, for the AF2 models, the method section is very brief yet the results states that “analyzed several parameters of the AF2 models to assess their ability to discriminate between correct and incorrect SLiM positions” but how these analyses were performed is not described.

3. Limited result presentation. For example:

a. Instead of showing a few select examples to claim model accuracy, what does the overall data show?

b. The results state a global accuracy metric was calculated and that the percent accuracy was 74% for correct models and 63% for incorrect models. How this was performed is not described and if there is a statistical difference between the correct and incorrect models is also not mentioned. This must be improved.

c. The figure legends need to provide more detail to describe what is shown. If an example of a SLIM is shown, include the reg ex pattern.

4. More detail on the comparison with AlphaMissense is needed. What dataset was used? How was it selected?

5. More context for the ACMG standards and aims is needed.

**Have the authors made all data and (if applicable) computational code underlying the findings in their manuscript fully available?**

Reviewer #1: **No: ** The pipeline is available through a Zenodo link, but the manuscript does not provide clear instructions on how users can access or implement MotSASi. Providing a public GitHub repository with all scripts and comprehensive documentation would significantly enhance its usability.

Reviewer #2: Yes

Reviewer #3: **No: ** Data and scripts are listed as a Zenodo file on the submission cover but not within the manuscript. Critical data that belong in the manuscript at least as summary tables in the supplementary material are missing.

PLOS authors have the option to publish the peer review history of their article (what does this mean? ). If published, this will include your full peer review and any attached files.

**Do you want your identity to be public for this peer review?** For information about this choice, including consent withdrawal, please see our Privacy Policy .

Reviewer #1: **Yes: ** Tamas Korcsmaros

Reviewer #2: No

Reviewer #3: No

**Figure resubmission:**

**Reproducibility:**



---

## [Decision Letter · Decision Letter 1]

13 May 2025

PCOMPBIOL-D-25-00157R1

Integrating AlphaFold2 Models and Clinical Data to Improve the Assessment of Short Linear Motifs (SLiMs) and Their Variants’ Pathogenicity

PLOS Computational Biology

Dear Dr. Brunello,

Thank you for submitting your manuscript to PLOS Computational Biology. After careful consideration, we feel that it has merit but does not fully meet PLOS Computational Biology's publication criteria as it currently stands. Therefore, we invite you to submit a revised version of the manuscript that addresses the points raised during the review process.

Please submit your revised manuscript within 60 days (July 13, 2025 at 11:59 PM). If you will need more time than this to complete your revisions, please reply to this message or contact the journal office at ploscompbiol@plos.org. Please include the following items when submitting your revised manuscript:

We look forward to receiving your revised manuscript.

Kind regards,

Mohammad Sadegh Taghizadeh, Ph.D.

Academic Editor

PLOS Computational Biology

Ilya Ioshikhes

Section Editor

PLOS Computational Biology

**Additional Editor Comments :**

Dear authors,

The esteemed reviewer #3 states that the manuscript is not well revised and her/his concerns have not been addressed. Also, the esteemed reviewer #2 has requested additional revisions. Therefore, I request that all the concerns be addressed point by point in the next step, and I sincerely inform you that if it is not so, I will reject this manuscript.

With the utmost respect

**Journal Requirements:**

1) We note that your MotSASi2.0_Revised_Manuscript files are duplicated on your submission. Please remove any unnecessary or old files from your revision, and make sure that only those relevant to the current version of the manuscript are included.

**Reviewers' comments:**

Reviewer's Responses to Questions

Reviewer #1: The authors properly addressed all the comments and made the github repo available.

Reviewer #2: The authors have thoroughly worked on the manuscript and addressed most of my comments. One remaining issue is the statement that, in clinical assessment, false negatives would be less concerning than false positives. Missing true positives clearly is a problem, especially if diagnosis is actionable (e.g. BRCA variants). That said, given that computational results alone will currently not be used for diagnosing, perhaps this point is not as critical. I would nevertheless recommend phrasing impacts of false positive/negative carefully - for any given patient, this is a massive impact on their quality of life.

I would like to include this paper: https://doi.org/10.1038/s44320-023-00005-6 which I believe would also be relevant for the authors to consider.

Reviewer #3: I don't find the improvements of this ms sufficient. There is still a disconnect between the methods and the results. The results are incomplete. It is briefly mentioned that most SLIMs are in disordered loops and perhaps why this methods outperforms AlphaMissense. Can the authors better describe what characterizes the SLIMs in their data and what characterizes accurately and inaccurately predicted SLIMs for the two methods? How do GO terms help the prediction?

The authors state "Our pipeline significantly expanded the repertoire of known functionally" but since these are predictions only and not functional assays, this must be corrected.

**Have the authors made all data and (if applicable) computational code underlying the findings in their manuscript fully available?**

Reviewer #1: Yes

Reviewer #2: Yes

Reviewer #3: **No: ** The results are incomplete

PLOS authors have the option to publish the peer review history of their article (what does this mean? ). If published, this will include your full peer review and any attached files.

**Do you want your identity to be public for this peer review?** For information about this choice, including consent withdrawal, please see our Privacy Policy .

Reviewer #1: **Yes: ** Tamas Korcsmaros

Reviewer #2: No

Reviewer #3: No

**Figure resubmission:**

**Reproducibility:**



---

## [Decision Letter · Decision Letter 2]

11 Jul 2025

Dear Dr. Brunello,

We are pleased to inform you that your manuscript 'Integrating AlphaFold2 Models and Clinical Data to Improve the Assessment of Short Linear Motifs (SLiMs) and Their Variants’ Pathogenicity' has been provisionally accepted for publication in PLOS Computational Biology.

Best regards,

Mohammad Sadegh Taghizadeh, Ph.D.

Academic Editor

PLOS Computational Biology

Ilya Ioshikhes

Section Editor

PLOS Computational Biology

Reviewer's Responses to Questions

**Comments to the Authors:**

Reviewer #2: The authors have addressed all my concerns.

Reviewer #4: The paper describes an application of the previously published method (MotSASi) for assessment of SLIMS using structures from AlphaFold2. The major novelty is incorporation of predicted structures instead of experimental ones.

The authors addressed the comments from previous review rounds.

**Have the authors made all data and (if applicable) computational code underlying the findings in their manuscript fully available?**

Reviewer #2: None

Reviewer #4: Yes

PLOS authors have the option to publish the peer review history of their article (what does this mean? ). If published, this will include your full peer review and any attached files.

**Do you want your identity to be public for this peer review?** For information about this choice, including consent withdrawal, please see our Privacy Policy .

Reviewer #2: No

Reviewer #4: No

---

## [Editor Report · Acceptance letter]

PCOMPBIOL-D-25-00157R2

Integrating AlphaFold2 Models and Clinical Data to Improve the Assessment of Short Linear Motifs (SLiMs) and Their Variants’ Pathogenicity

Dear Dr Brunello,

I am pleased to inform you that your manuscript has been formally accepted for publication in PLOS Computational Biology. Your manuscript is now with our production department and you will be notified of the publication date in due course.

With kind regards,

Livia Horvath
